# The Role of EEG in the Diagnosis, Prognosis and Clinical Correlations of Dementia with Lewy Bodies—A Systematic Review

**DOI:** 10.3390/diagnostics10090616

**Published:** 2020-08-20

**Authors:** Zhe Kang Law, Carein Todd, Ramtin Mehraram, Julia Schumacher, Mark R. Baker, Fiona E. N. LeBeau, Alison Yarnall, Marco Onofrj, Laura Bonanni, Alan Thomas, John-Paul Taylor

**Affiliations:** 1Translational and Clinical Research Institute, Biomedical Research Building, Campus for Ageing and Vitality, Newcastle University, Newcastle upon Tyne NE4 5PL, UK; zhe-kang.law@ncl.ac.uk (Z.K.L.); carein.todd@ncl.ac.uk (C.T.); r.mehraram2@ncl.ac.uk (R.M.); julia.schumacher@ncl.ac.uk (J.S.); mark.baker@ncl.ac.uk (M.R.B.); alison.yarnall@ncl.ac.uk (A.Y.); alan.thomas@ncl.ac.uk (A.T.); 2Department of Medicine, National University of Malaysia, Kuala Lumpur 56000, Malaysia; 3Department of Clinical Neurophysiology, Royal Victoria Infirmary, Queen Victoria Rd, Newcastle upon Tyne NE1 4LP, UK; 4Biosciences Institute, Newcastle University, Faculty of Medical Sciences, Framlington Place, Newcastle upon Tyne NE2 4HH, UK; fiona.lebeau@ncl.ac.uk; 5Department of Neuroscience Imaging and Clinical Sciences and CESI, University G D’Annunzio of Chieti-Pescara, 66100 Chieti, Italy; onofrj@unich.it (M.O.); l.bonanni@unich.it (L.B.)

**Keywords:** dementia with lewy bodies, lewy body disease, parkinson’s disease dementia, electroencephalography, electrophysiology, systematic review

## Abstract

Despite improvements in diagnostic criteria for dementia with Lewy bodies (DLB), the ability to discriminate DLB from Alzheimer’s disease (AD) and other dementias remains suboptimal. Electroencephalography (EEG) is currently a supportive biomarker in the diagnosis of DLB. We performed a systematic review to better clarify the diagnostic and prognostic role of EEG in DLB and define the clinical correlates of various EEG features described in DLB. MEDLINE, EMBASE, and PsycINFO were searched using search strategies for relevant articles up to 6 August 2020. We included 43 studies comparing EEG in DLB with other diagnoses, 42 of them included a comparison of DLB with AD, 10 studies compared DLB with Parkinson’s disease dementia, and 6 studies compared DLB with other dementias. The studies were visual EEG assessment (6), quantitative EEG (35) and event-related potential studies (2). The most consistent observation was the slowing of the dominant EEG rhythm (<8 Hz) assessed visually or through quantitative EEG, which was observed in ~90% of patients with DLB and only ~10% of patients with AD. Other findings based on qualitative rating, spectral power analyses, connectivity, microstate and machine learning algorithms were largely heterogenous due to differences in study design, EEG acquisition, preprocessing and analysis. EEG protocols should be standardized to allow replication and validation of promising EEG features as potential biomarkers in DLB.

## 1. Introduction

Dementia with Lewy bodies (DLB) is a common late life dementia with 25–45% of dementia cases demonstrating Lewy body pathology at autopsy [1,2]. DLB is associated with significant morbidity [3]. In addition to cognitive impairment it presents with a multiplicity of symptoms ranging from psychiatric, sleep, motor and autonomic manifestations; this complexity can make management challenging [4]. Furthermore, the diagnosis of DLB remains suboptimal with many patients misdiagnosed as AD, even with improved diagnostic criteria [5,6].

Investigations such as dopamine transporter imaging, polysomnography, ^123^iodine-Metaiodobenzylguanidine (^123^iodine-MIBG) myocardial scintigraphy have high specificity and sensitivity and have been designated as indicative biomarkers for the diagnosis of probable DLB [6]. In addition, a number of ancillary biomarkers for the diagnosis of DLB have been suggested. One of these is electroencephalography (EEG). Specifically, posterior slow-wave activity with periodic fluctuations in the pre-alpha/theta range on resting EEG was designated as a supportive biomarker in the 2017 DLB consortium diagnostic criteria [6]. Theta activities had been correlated with cognitive decline, cognitive fluctuations and hallucinations [7]. The clinical correlation of EEG enables it to be used as a biomarker for more difficult to evaluate clinical symptoms such as cognitive fluctuations. Furthermore a key area, given the potential for early intervention and disease modification, is the diagnosis of prodromal DLB [8,9] and a key question would be whether EEG has any diagnostic utility at this early stage.

However, to date, the value of EEG and its use in DLB diagnosis and management has been limited. EEG abnormalities are often non-specific to DLB; slowing of EEG activities may be caused by other neurodegenerative diseases, toxic or metabolic encephalopathies, central nervous system infections, sedative medications as well as somnolence [10]. Furthermore, previously EEG studies in DLB were largely preliminary with substantial heterogeneity across studies [11]. The advent of more advanced computational approaches to quantify EEG changes as well as larger multicentre studies may allow for improved diagnostic sensitivity and specificity.

Beyond diagnosis, EEG may also have a role in prognostication and the prediction of treatment response. Treating cholinergic dysfunction in DLB with cholinesterase inhibitors may improve global cognitive function, cognitive fluctuations, hallucinations and activities of daily living [4]. However, the response to cholinesterase inhibitors is variable with only half of patients benefiting [12,13,14]. Quantitative EEG, as a measure of electro-cortical activity may be sensitive to measuring cholinergic function and consequently pharmacological intervention [15]. Changes in the EEG from slow delta wave (non-aroused state) to fast beta and gamma (aroused state) as well as alpha reactivity to eyes opening are strongly associated with the activation of cortical cholinergic projections from the nucleus basalis of Meynert (NBM) [16,17,18].

In this systematic review, we aim to provide an up-to-date appraisal of the literature pertaining to the use of EEG in DLB and address the following: (1) can EEG be used to differentiate DLB from other dementias?; (2) what are clinical correlates of EEG in DLB?; (3) can EEG be used to predict response to cholinesterase inhibitors?; and (4) can EEG be used to diagnose prodromal DLB and predict conversion to dementia?

## 2. Methods

### 2.1. Search Strategy and Selection Criteria

We performed this study in accordance with the Preferred Reporting Items for Systematic Reviews and Meta-Analyses (PRISMA) statement [19] (Appendix A). MEDLINE, EMBASE, and PsycINFO were searched using search strategies ( Appendix A) for potentially relevant articles up to 6 August 2020. Reference of relevant review articles and systematic reviews were hand searched for potential studies as well.

The selection criteria were intentionally broad. We included observational studies (any design) that enrolled participants with DLB diagnosed using the 1996 [20], 2005 [21] or 2017 [6] DLB consensus criteria, or postmortem studies. Studies that reported DLB and Parkinson’s disease dementia (PDD) as a combined group (termed Lewy body dementia; LBD), were included but studies that only included Parkinson’s disease (PD) with or without cognitive impairment were excluded. We also included studies that recruited participants with mild cognitive impairment with Lewy bodies (MCI-LB) [9] to explore the role of EEG in predicting disease progression. Studies that reported results of resting state EEG and event-related potentials (ERPs) were included but studies of local field potentials were excluded given that the latter recordings can only be made with invasive depth electrodes and are, therefore, unlikely to become a routine clinical biomarker. Only full text article publications in English were included.

### 2.2. Selection of Studies and Data Extraction

The eligibility of the studies was first assessed by screening titles and abstracts. Full text articles of potentially eligible studies were then screened for possible inclusion. Two authors (ZKL and CT) performed the selection process independently with the conflict resolved by referring to a third author (JPT). Two authors (ZKL and CT) independently extracted data using a data extraction form (Appendix A). Data extracted included author, year published, design, diagnosis, clinical symptoms, EEG acquisition/setup and EEG features used.

### 2.3. Assessment of Bias and Quality

Risk of bias was assessed using the 10-point checklist for case series developed by the Joanna Brigg Institute (JBI) [22,23], where a score of ≤6 was considered low quality, 7–8 moderate and 9–10 high quality. In addition, we assessed possible bias/confounders in the acquisition of EEG data (7 point checklist: reporting of EEG acquisition; ensuring an awake recording; concurrent psychoactive medication; exclusion of artefacts; age-matching; cognitive scores-matching; and blinding of assessors/engineers, Appendix A). An EEG checklist score of ≤3 indicated low quality, 4–5 moderate and 6–7 high quality.

### 2.4. Statistical Analysis

Due to the heterogeneity of included studies, we did not perform a meta-analysis but provide a description of included studies. In interpretation of study reports, *p* value of <0.05 was considered statistically significant. Individual studies’ sensitivity, specificity, overall accuracy and area under receiver operating characteristics curve were described if reported in the included studies. Correlation coefficients were interpreted as follows: 0 = negligible correlation; 0.1 to 0.3 = weak; 0.4 to 0.6 = moderate; 0.7 to 0.9 = strong; and 1 = perfect [24].

## 3. Results

The initial search yielded 1264 studies and after title and abstract screening 655 studies were excluded and 220 full text articles were examined (Figure 1). The number of included studies amounted to 43 with 177 exclusions for various reasons (Figure 1).

### 3.1. Characteristics of Included Studies

These studies were conducted between 1999 and 2020, with a total of 6901 participants of which 1382 (20.0%) were DLB patients. There were 21 single-centre and 22 multi-centre studies. However, some of the studies were conducted on the same cohort/patient group. With the exception of one study with post-mortem confirmation [25], the diagnosis of DLB was made based on clinical criteria (Table 1).With the exception of one study that involved patients with severe DLB and moderate AD, all studies involved patients with mild (26 studies, Mini-Mental State Examination [MMSE] ≥ 20) to moderate dementia (4 studies, MMSE ≥ 10–20) or mild cognitive impairment (MCI). Six studies did not explicitly report cognitive scores.

### 3.2. Electroencephalography (EEG) Acquisition

Thirty-two studies used the 10-20 EEG placement system with 14 to 30 scalp electrodes/channels, seven studies using the 10-5 system with 128 electrodes and two studies combined data acquired using 10-20 and 10-5 system. One study used only four electrodes to study ERPs and one study did not specify the EEG placement system. Fourteen studies recorded EEG for ≥20 min, 13 studies recorded between 2 to 10 min and the duration of recording was not specified in 16 studies. A wide range of makes of EEG machines were used with 11 different systems described across the included studies. The frequency band definitions used in the studies were largely not standardized with various definitions used (Appendix A).

### 3.3. Potential Confounders

Thirty studies reported exclusion of EEG epochs with artefacts while this was not reported in seven; in six studies this was not relevant as they were qualitative studies (Appendix A). Twenty-eight studies (65%) described procedures to ensure patients were awake during the recording whereas 15 studies did not explicitly report them. Nineteen studies (44.2%) reported blinding of EEG assessors/technicians to diagnosis, one study (2.3%) reported no blinding while 23 (53.5%) did not report whether blinding occurred. In 34 studies, participant groups were age-matched, in six studies, participants were not, and in three studies it was unclear. Apart from healthy controls, cognitive scores were matched between dementia groups in 28 studies and unmatched in 11 studies, including two where the studies aimed to explore EEG across different stages of cognitive impairment. In four studies, it was unclear if the cognitive scores were matched. Seven studies excluded participants with psychotropic drugs from analyses, 23 did not and in 13 studies it was unclear if they were excluded. Amongst the 23 studies that included participants taking psychotropic drugs, the proportion of participants taking them were different between dementia groups (DLB, PDD, AD, and other dementias) in 16 studies. In all but one of these studies, a higher proportion of DLB patients had psychotropic medication compared to AD. In one study which compared PDD to DLB, more PDD than DLB patients were taking L-dopa. In five studies, the proportions of psychotropic drug users were unclear.

### 3.4. Quality of Included Studies

Of the 43 studies included, four studies were classified overall as high quality, 35 were moderate and four low quality based on JBI study quality and EEG checklist (Table 1). In the JBI study quality checklist, one study was assessed as high quality, 27 moderate and 15 low quality. On the EEG checklist, 10 studies were assessed as high, 21 moderate and 12 low quality (Appendix A).

### 3.5. EEG Analysis

Figure 2 illustrates the EEG features and methods used in the studies. These were classified on the basis of the predominant analysis employed: qualitative, spectral power, connectivity, and/or microstates analysis. In addition, we also included machine-learning approaches which utilised a combination of these analyses/features to differentiate between dementia subtypes. Definitions of EEG features are described in Appendix A.

### 3.6. EEG for Comparison of Dementia with Lewy Bodies (DLB) vs. Alzheimer’s Disease (AD)/Other Diagnoses

Forty-three studies compared EEG of DLB with other diagnoses, 42 of them included comparison of DLB with AD, 10 studies compared DLB with PDD and 6 studies compared DLB with other dementias (vascular dementia, frontotemporal lobar degeneration) and depression. One study made comparison between DLB and healthy controls only. Thirty studies included healthy participants as a control group (Table 1).

#### 3.6.1. DLB vs. AD

##### Visual EEG Assessments

Ten studies included visual EEG assessments as part of the study assessments (Table 1) [25,26,27,28,29,47,49,50,57]. The majority of DLB patients (up to 97%) in these studies showed focal and diffuse EEG abnormalities [25,50,57]. More than 90% of DLB patients had theta/delta activity in the posterior, anterior/temporal regions and only 5% had normal alpha activity compared to AD patients where 10% have theta/delta and 90% alpha activity [50].

Three studies classified the severity of EEG abnormalities based on a 4- or 5-point scale i.e., 1 = normal EEG; 2 = mildly abnormal; 3 = moderately abnormal; 4 = severely abnormal with an addition of 5 = isoelectric in one study. Some studies found EEG severity scores to be significantly worse in DLB than in AD [50] while others showed no difference [28,29]. However, the features used to assess severity were not standardised and often not clearly defined [28,29,50].

Two studies used the grand total EEG (GTE) score, a standardized scoring system, in their visual assessments. The GTE score graded EEG based on background rhythm frequency, reactivity, diffuse slow wave, focal and paroxysmal activities, with a score of 2 to 31, with higher scores indicating more severe EEG abnormalities. The GTE score’s sensitivity, specificity and area under receiver operating characteristics curve (AUROC) in differentiating probable DLB from AD were 72%, 85% and 0.9; and 79%, 76% and 0.78 at cut-off of 9.5 [26] and 6.5 [27], respectively. In particular, frontal intermittent rhythmic delta activities (FIRDA) was commonly found in DLB (17.2–33.3%) and less frequent in AD (1.8–5.6%) [26,27,49,50]. The adjusted odds ratio (aOR) for diagnosis of probable DLB (compared to AD) was 11.0–27.7 [26,27] when FIRDA is present.

##### Quantitative EEG

Spectral Power Analyses

The dominant background rhythm frequency is consistently reported to be lower in DLB compared to AD (using various measures: mean peak/dominant frequency). The mean peak/dominant frequency ranged between 6.7–7.5 Hz for DLB and 7.5–8.8 Hz for AD [33,34,35,37,38,46,47,49,50,64]. Several studies which used the compressed spectral arrays (CSA) method found that between 95–100% of patients with DLB had dominant frequency of <8 Hz while 85–90% of patients with AD had dominant frequency of >8 Hz [34,35,37,40]. This agrees with findings from visual EEG assessments where a dominant rhythm of <8 Hz would be classified as abnormal and supportive of DLB diagnosis. This consistently translates into higher relative power in lower frequency bands and lower relative power in higher frequency bands and a higher theta/delta to alpha/beta ratio [33,34,35,37,38,46,47,49,50,64] (Table 2).

To account for inter-individual variability, some studies use individual alpha frequency peaks (IAF) and transition frequency (TF). IAF is defined as the maximum power density peak in the alpha range (6–14 Hz) [51]. TF marks the transition frequency between alpha and theta, defined as the minimum power density between 3 to 8 Hz [51]. Based on TF and IAF, the frequency bands for each subject were estimated as follows: delta from TF-4 Hz to TF-2 Hz, theta from TF-2 Hz to TF; low-frequency alpha band (alpha 1 and alpha 2) from TF to IAF; and high-frequency alpha band (or alpha 3) [51]. Mean IAF (equivalent of dominant/peak frequency) and TF were reduced in DLB compared to AD [51,63] even at the mild cognitive impairment (MCI) stage [53]. On spectral analysis, alpha relative power in occipital regions was reduced in AD compared to DLB while delta relative power was higher in DLB than AD [51,53]. Higher alpha power and lower delta power differentiate AD from DLB with sensitivity and specificity of 65–78% in multicentre studies [51,53].

Tanaka et al. introduced the concept of neuronal activity topography (NAT), a measure of brain topography based on spectral power which indicates the level of activity and synchrony. This was able differentiate DLB from healthy controls in 86% of patients [56].

The findings of EEG variability in DLB have not been consistent. The variability of mean dominant frequency is greater in DLB than AD in some studies [31,34,35,37] whereas others have shown greater variability in AD [38,46,50]. Additionally, delta power variability was reportedly greater in DLB compared to AD in two studies [30,33].

Bonanni et al. combined frequency and variability measures and classified them according to five combined spectral array (CSA) patterns (Appendix A) [34]. A CSA pattern of stable alpha (dominant alpha in ≥60% of epochs) was present in 0% of DLB and 100% of AD patients [34]. However, subsequent studies suggest that CSA pattern >2 may better classify DLB compared to AD [37]. Different DFV cut-offs of >1.2 Hz [34,40] and >2.2 Hz [37] combined with other parameters (frequency prevalence of alpha/pre-alpha) have been used to differentiate DLB from AD with an accuracy ranging from 75% to 100%.

One study explored the EEG changes in DLB and AD as the disease progressed from mild dementia (mean MMSE 22.3 and 22.8 respectively) to the moderate stage (mean MMSE 17.1 and 17.2 respectively) after a 2-year follow-up [34]. While there were decline in dominant frequency in DLB (from 7.4 to 6.8 Hz) and AD (from 8.3 to 8.0 Hz), the dominant rhythm remained within normal alpha range in 90% of AD while they were theta/delta range in 94% of DLB [34].

In studies that reported spectral power analyses according to brain regions, reduction in dominant frequency with increase delta power and reduced alpha power affected all posterior, temporal, central and anterior regions [32,34,37,38,40,51]. Increased theta/delta power or activities appeared to be more prominent in the posterior region in DLB patients in several studies [32,34,51]. On the other hand, the dominant frequencies were more reduced in the anterior region (5.9–7.0 Hz) compared to posterior (6.9–7.4 Hz) in DLB patients in several studies [34,37,64]. However, in these studies, AD patients’ dominant frequencies were lower in the anterior region (7.3–8.4 Hz) than posteriorly (8.3–8.8) as well. Although the dominant frequency was lower with more pre-alpha activities in the anterior region, the diagnostic accuracy of posterior pre-alpha rhythm was higher in differentiating DLB from AD [34,36,37]. Contrastingly another study reported that pre-alpha activities were more prevalent in the anterior region (88%) compared to posterior (74%) in DLB and the presence of anterior pre-alpha with posterior alpha appeared specific to DLB when compared to AD and healthy controls [64].

Connectivity

Twenty studies investigated connectivity including coherence, Granger causality, phase lag index (PLI), weighted PLI, lagged linear connectivity (LLC), and global field synchronisation (GFS). Coherence is the correlation between signals x and y as a function of the frequency, ranging between 0 and 1 [68]. This measure may be influenced by volume conduction through the scalp. Coherence in the theta range, was reported in two studies to be higher in DLB compared to AD [32,33]. On the other hand, coherence in the alpha and beta range was found to be reduced in DLB compared to AD in one study [33], while others have reported higher alpha and beta coherence [32] in DLB.

Granger causality was used to describe whether the time course of the EEG in channel X can help to predict the future values of the EEG signal in channel Y [69]. One study found parietal region Granger causality to be significantly higher in DLB compared to AD and have a high accuracy of ~100% [41].

The PLI estimates consistent causal delay between two signal sources and is less affected by the scalp’s volume conduction. PLI scores are bounded between 0 and 1, where 0 means lack of causal synchronisation and 1 full causal synchronization [70]. The PLI within the alpha range was reportedly lower in DLB than AD, indicating more severe changes in connectivity in DLB [45,49,50]. The alterations in alpha network connectivity resonate with another study which reported reduced mean alpha band directed phase transfer entropy (dPTE), which measures posterior-to-anterior connectivity, in DLB compared to AD [48].

Weighted PLI is a development of PLI which involves weighting the PLI values with the imaginary part of the cross-spectrum between the two time-series [71]; the latter part of the cross spectrum is associated with the phase difference, i.e., the delay, between the signals. An imaginary part close to zero means that the two signals are almost overlapping. One benefit of weighted PLI is that PLI may be artificially increased by noisy conducting sources, whilst in weighted PLI this effect tends to be lower [71]. Only one study used this approach, finding that weighted PLI was lower in the beta band in DLB compared to AD [61].

LLC is a connectivity measure that is obtained using exact low-resolution brain electromagnetic tomography (eLORETA) software. LLC estimates functional cortical source connectivity removing zero-lag instantaneous phase coupling between cortical sources of resting state EEG rhythms and is less influenced by volume conduction [72]. LLC in alpha and delta ranges was reduced in AD compared to DLB, which Babiloni et al. suggested might imply that there was more cortical disconnection in AD than in DLB as both conditions progress to dementia [52].

A graph theory approach based on weighted network and minimum spanning tree (MST) analyses has also been implemented to assess functional network connectivity. The one study which examined weighted PLI demonstrated weaker connectivity and greater network segregation in the beta network in DLB compared to AD [61]. MST was used in four studies [45,46,47,50] which reported lower degree, lower betweenness centrality, higher diameter, higher eccentricity and lower leaf-fraction in DLB compared to AD [45,50] indicating loss of hubness and a less-efficient network. The MST in DLB appears to have a randomised pattern associated with decreased efficiency and reduced synchronization compared to AD [46]. Table 3 summarises EEG connectivity findings in DLB.

Microstate

EEG microstates are transiently stable brain topographies whose temporal characteristics provide insight into the brain’s dynamic repertoire. In one study, microstate duration was found to be increased with a reduced number of microstates per second in patients with LBD compared to AD and healthy controls [55]. Longer microstate duration was correlated with a loss of dynamic functional connectivity between the basal ganglia and thalamic networks and large-scale cortical networks on functional magnetic resonance imaging (fMRI) [55].

Reactivity

Reactivity of rhythmic background activity to eyes opening was reduced in DLB compared to AD in qualitative and quantitative analyses in one study [26]. More recently, Schumacher et al. [63] showed that alpha reactivity, quantified by the relative reduction in alpha power over occipital electrodes when opening the eyes, was reduced in DLB and PDD patients compared to AD and healthy controls. The loss of reactivity correlated with loss of NBM volume particularly in PDD patients.

Event-Related Potential

Delayed P300 and anterior-to-posterior scalp amplitude gradient inversion on an auditory oddball paradigm evoked potential can differentiate DLB from AD [35]. Pre-pulse inhibition, a marker of auditory sensory filtering, was shown to be reduced in DLB compared to AD and healthy controls [59]. Visual P3 was significantly delayed in DLB compared to AD [58]. However, evoked response to visual stimuli appears to be more affected than responses to auditory stimuli in DLB patients, with the Visual P3/Auditory P3 latency ratio (VP3/AP3, comparing the prolongation of latency with visual and auditory evoked response) significantly higher in DLB than in AD [58].

Machine-Learning Algorithms

Machine-learning algorithms, using between 2 to 61 features, were utilised in seven studies to differentiate DLB and AD. The accuracy of machine learning methods ranged between 66% to ~100% [39,41,42,43,44,47,50,61]. The number of features used, however, does not appear to influence accuracy. One study found that only including just two features, Granger causality and the ratio of theta band power and beta1 band power in the model, accuracy, sensitivity and specificity reach ~100% [41].

Multimodal Use of EEG and Other Biomarkers

Several studies have combined the use of neuroimaging, CSF, and key clinical features with EEG. In these studies EEG showed better accuracy than neuroimaging and CSF biomarkers in diagnosing DLB compared to AD [43,44,47]. The combination of EEG and structural magnetic resonance imaging (MRI) also may increase accuracy, sensitivity and specificity to >90% [44] in differentiating DLB from AD.

#### 3.6.2. Prodromal DLB and Progression to Dementia

EEG abnormalities on visual rating have been reported to be more common in DLB even at the mild cognitive impairment (MCI) stage. Comparing MCI with Lewy bodies (MCI-LB) and MCI due to AD (MCI-AD), diffuse abnormalities (76% vs. 8%) and FIRDA (22% vs. 0%) were more common in MCI-LB [66]. EEG severity scores were significantly worse in MCI-LB as well, where only 16% of MCI-LB had normal EEG while EEGs were normal in 49% of MCI-AD [66].

Quantitative EEG findings of MCI-LB have been reported to be similar to those reported in DLB vs. AD, where MCI-LB have lower dominant frequency compared to MCI-AD [65,66,67]. This also translates into higher pre-alpha power and lower alpha and beta power and a higher theta/alpha ratio [65,66,67]. Schumacher 2020b reported that spectral power analyses had sensitivities of 23–51%, specificity of 81–97% and area under receiver operating characteristics curve (AUROC) of 0.54–0.71 in diagnosing MCI-LB (Table 4) [67]. On the other hand, van der Zande 2020 reported AUROC of 0.76–0.97 but sensitivities and specificities were not reported [66]. Connectivity of MCI-LB was only studied in one study, which reported that LLC within the alpha range, were decreased in MCI-LB and MCI-AD compared to age-matched controls; however there was no difference between the MCI groups [54].

Several studies explored whether EEG features can predict progression to dementia in patients with MCI. One study reported a reduction in mean frequency and alpha/theta ratio in MCI who progressed to DLB (MCI-LB) compared to those who progressed to AD(MCI-AD) [65]. Another study used CSA to predict progression in patients with MCI to DLB, AD or no progression at 3 years with an overall accuracy of 76%. In this study all patients with MCI who progressed to DLB had a CSA pattern of >1 (1 plus to 5) while 93% who progressed to AD had a CSA pattern 1 (stable alpha) at baseline [36]. In comparison, the presence of one or more core or supportive clinical features of DLB predicted progression to DLB in 75% of patients with MCI. When MCI patients who progressed to DLB (MCI-LB) were compared to DLB patients, the dominant frequency variability was similar although DLB patients had lower mean dominant frequencies [36]. Although both MCI-LB and MCI-AD group had similar MMSE at baseline (mean 25.7 and 25.9 respectively) and a similar decline on follow-up (20.6 in DLB and 20.5 in AD), follow-up EEG of patients with MCI-DLB showed progression where all patients with CSA 1 plus progressed to CSA 2 or 3. Conversely, follow-up EEG of patients with MCI-AD showed no progression (93% with CSA pattern 1) despite the cognitive decline.

More recently, van der Zande et al. identified several EEG features that predicted shorter progression to dementia stage in patients with MCI-LB including diffuse abnormalities, defined as dominant frequency <8 Hz affecting all brain regions, (hazard ratio, HR 9.9, 1.9–49.3), EEG severity scores of >2 (HR 4.1, 95%CI 1.4–11.3) and lower alpha-2 power of <0.06 (HR 5.1, 1.5–16.5) [66]. The mean time to dementia was 2.8 years in patients with diffuse abnormalities compared to 6.9 years when this is absent [66].

#### 3.6.3. Summary of EEG Accuracy

In summary, the accuracy of EEG features in differentiating DLB from AD, in the MCI to moderate dementia stage, varies between fair to excellent, depending on type and number of features used (Table 4).

#### 3.6.4. DLB vs. Parkinson’s Disease Dementia (PDD)

EEG findings of DLB and PDD with cognitive fluctuations appears similar while differences have been observed between DLB and PDD without cognitive fluctuations. DLB and PDD with cognitive fluctuations tend to have lower mean frequency, higher theta/delta and lower alpha power as well as lower alpha/theta ratios than PDD without cognitive fluctuations and AD [34,38,46,51,52]. In the study by Bonanni et al. a stable alpha CSA pattern was not found in any patients with DLB and PDD with cognitive fluctuations [34]. In all DLB and PDD patients with cognitive fluctuations the prevalence of a pre-alpha frequency was >40% and the prevalence of an alpha frequency was <32%, while in all PDD patients with no cognitive fluctuations, the prevalence were ≤ 11% and ≥55% respectively [34]. A reduction in functional connectivity (LLC and GFS) in the alpha band has also been noted in DLB compared to PDD [52,60].

Pre-pulse inhibition (auditory sensory filtering) appears moderately reduced in PDD compared to AD, but is less affected when compared to DLB [59]. Both DLB and PDD have reduced reactivity to eyes opening [60,63] and intermittent photic stimulation [60].

#### 3.6.5. Mixed DLB/AD

Patients with a clinical diagnosis of AD with visual hallucinations share certain EEG features with DLB including lower dominant frequency, higher theta, lower alpha power, and increased theta/alpha ratio when compared to AD patients without hallucinations [49]. AD patients with hallucinations had higher alpha band connectivity, as measured with PLI, compared to DLB, but lower connectivity compared to AD with no hallucinations. This suggests that the severity of EEG abnormalities in AD patients with hallucinations lies between DLB and AD without hallucinations [49]. Similar findings were also observed in DLB patients with CSF tau/Aβ-42 ratio > 0.42 (i.e., likely mixed DLB/AD disease). Patients with mixed DLB/AD disease appear to have lower dominant frequency, higher theta power, lower alpha power, increased theta/alpha power ratio, and lower PLI compared to AD, whereas no differences have been observed between DLB and mixed DLB/AD disease [50].

#### 3.6.6. DLB vs. Depression/Psychiatric Disorders

In one study that compared visual EEG assessment of DLB and psychiatric disorders, a normal EEG was only present in 1.2% of patients with DLB with 6.2% having only focal abnormalities, 30.9% having diffuse (only) abnormalities and 61.7% both focal and diffuse abnormalities [57]. Conversely, 60% of patients with psychiatric disorder had normal EEG, 29% had only focal abnormalities, 16.1% only diffuse abnormalities and 13% both focal and diffuse abnormalities. However, the types of medication taken by both groups of patients were not reported [57]. Two studies that used machine-learning algorithms including a combination of relative and absolute powers of delta to gamma frequency band, peak frequency, power ratio and coherences, found a AUROC of 0.95 to 0.98 in differentiating DLB from depression [39,42].

### 3.7. Clinical Correlates of EEG Measures in DLB

#### 3.7.1. Cognitive Fluctuations

Four studies described the correlations of EEG findings with cognitive fluctuations. One qualitative study reported temporal slow-wave transients to be significantly associated with episodes of loss of consciousness [25]. Frequency variability of the theta and delta band in the central, posterior, and lateral regions correlated significantly with cognitive fluctuation scores (either Cognitive Drug Research Computerised Assessment System-Dementia, COGDRAS-D and Clinician Assessment of Fluctuation, CAF) with a Spearman’s coefficient (*r*) of 0.42 to 0.81 [30,31,38]. In addition, P300 latency (Pearson’s correlation coefficient, rs = 0.6), amplitude (rs = −0.5) and anterior-to-posterior amplitude distribution gradient correlated with CAF (rs = 0.5) [35]. In microstates analysis, longer mean microstates duration correlated with worse cognitive fluctuations (Mayo fluctuation scale, rs = 0.56) [55].

#### 3.7.2. Hallucinations

Dauwan and colleagues [49] compared 3 groups of patients: DLB (all of whom had hallucinations), AD with visual hallucinations and AD without hallucinations. DLB patients had similar peak frequency compared to AD with hallucinations but lower frequency than AD without hallucinations. DLB and AD with hallucinations had lower alpha and higher theta power compared to AD without hallucinations. Alpha range PLI was also lower in DLB and AD with hallucinations compared to AD without hallucinations. These findings suggest that slowing of dominant rhythm and decreased functional connectivity is associated with the presence of hallucinations. In another study, memory perception network in the EEG-resting state network correlated with hallucination component of neuropsychiatric inventory (NPI) scale (rs = 0.44) [62].

As noted previously, patients with visual hallucinations (PDD with visual hallucinations and DLB) have higher VP3/AP3 latency ratio compared to those without hallucinations (AD and PDD without visual hallucinations) [58]. A cut-off VP3/AP3 latency ratio of >1.21 had moderate accuracy in differentiating patients with visual hallucinations from those without visual hallucinations (AUROC 0.73, accuracy 70%, sensitivity 69% and specificity 70%) [58].

#### 3.7.3. Degree of Cognitive Impairment

Severity of EEG abnormalities from visual assessment (rs = −0.61) [28], dominant frequency in anterior and posterior derivations (rs = 0.2–0.3), frequency prevalence (FP) theta in anterior and posterior derivations (rs = −0.2); FP alpha anteriorly (rs = 0.2), alpha (rs = 0.35–0.38) and delta power (rs = −0.25 to −0.32) [51,53], GFS (Pearson’s coefficient, *r* = 0.43) [60], P300 latency (rs = −0.6) [35] as well as lower leaf fraction (rs = 0.27), higher diameter (rs = −0.29) and higher eccentricity (rs = −0.34) on MST correlated with MMSE scores [45]. Another study noted that occipital alpha activity and memory perception network correlated with Alzheimer’s Disease Assessment Scale-Cognitive Subscale (ADAS-Cog) with rs = −0.37 and −0.32 respectively [62].

#### 3.7.4. Domains of Cognitive Impairment

Correlations of EEG features and specific domains of cognitive function in DLB were reported in five studies (Table 5), primarily fronto-executive and visual. The correlation coefficient values ranged between 0.29 to 0.60 indicating weak to moderate correlations.

#### 3.7.5. Neuropsychiatric Symptomatology

Weighted PLI in alpha and beta range correlated with the NPI hallucinations subscale in one study (rs = −0.61 for alpha; rs = −0.53 for beta) in DLB patients [61]. The visual perception network correlated with the anxiety (rs = −0.36) and depression (rs = −0.43) components of the NPI, whereas the self-referential network associated with NPI-depression (rs = −0.32) [62,64].

In one ERP study, P300 latency (rs = 0.6), anterior-to-posterior latency distribution gradient (rs = 0.5), amplitude (rs = −0.5), and anterior-to-posterior amplitude distribution gradient (rs = 0.5) correlated significantly with total NPI score in one study [35].

#### 3.7.6. Motor Symptoms

One study reported a reduced GFS response to intermittent photic stimulation with more severe motor symptoms in PD patients (Unified Parkinson’s Disease Rating Scale [UPDRS], *r* = −0.40, *p* = 0.04) but correlation in DLB was not reported [60].

### 3.8. Treatment Effect on EEG

The effect of cholinesterase inhibitors (ChEI) on EEG is unclear, with two studies showing no effect [26,27]. However, both studies had a small number of participants taking cholinesterase inhibitors with only two DLB and AD patients in one study [26] and one DLB and two AD patients in another [27] taking these agents. Both studies also did not have a before and after cholinesterase inhibitor EEG comparison. In another study, donepezil-treated DLB patients have been reported to have lower delta and theta power and coherence than those not treated with donepezil [32]. Treatment with dopaminergic agents did not have a significant effect on alpha power and reactivity among patients with DLB and PDD in one study [63].

## 4. Discussion and Conclusions

A main finding from our systematic review is that a normal EEG strongly argues against a diagnosis of DLB; >90% of DLB patients have diffuse abnormalities while an estimated 5–10% of AD patients have abnormal EEGs [50]. The most consistent EEG finding that differentiates DLB from AD appears to be the slowing of the background rhythm. The dominant background rhythm was predominantly within the alpha range in AD, whilst it tends to be lower (pre-alpha or high theta) in DLB. This is also reflected in quantitative EEG analyses, with the consensus from many studies being an increase in theta/delta power, a decrease in alpha/beta power, and consequently a reduced alpha/beta to theta/delta ratio. A dominant frequency cut-off of <8 Hz may differentiate DLB from AD in 85–100% of patients [34,37,40].

The slowing of dominant frequency in DLB compared to AD is apparent from MCI stage and persists as a differentiating feature in mild to moderate dementia stage. In corresponding stages of dementia with comparable MMSE scores, dominant frequencies are reported to be slower in DLB compared to AD. Our synthesis of the literature is in agreement with EEG studies in AD, where the dominant frequencies are reported within alpha range in the mild to moderate stage, with slowing occurring as the dementia progresses [73,74] although there has not been any comparison between DLB and AD, as yet, of EEG changes in the severe dementia stage.

The slowing of dominant frequency with its associated spectral power features had moderate (AUROC 0.54–0.71, sensitivities 23–51%, specificities 81–97%) to good (AUROC 0.76–0.97) performance in diagnosing MCI-LB [66,67]. Thus, potentially at the prodromal stage, EEG may be comparable to dopaminergic imaging which had a sensitivity of 61% and specificity of 89% in diagnosing MCI-LB [75]. In addition, EEG features including CSA pattern >1, diffuse slowing, EEG severity grade and alpha-2 power were predictive of progression to dementia [36,66]. However, given the small number of studies performed to date, variable accuracy performance, as well as lack of clarity on what is the optimal EEG spectral feature(s) for diagnostic separation, further validation studies in prodromal DLB groups are needed.

The slowing of dominant frequency affects all brain regions, although slowing in the posterior rhythm have a higher diagnostic yield in differentiating DLB from AD [34,36,37]. Notably, there was one study which showed that the appearance of anterior pre-alpha rhythms may occur and be specific to DLB in the mild stage [64] and thus this EEG feature could herald the emergence of dementia; however further evaluation in longitudinal cohorts, particularly from earlier, prodromal or MCI stages would be needed to support this assertion.

A normal alpha dominant posterior rhythm is thought to be a product of thalamo-cortical interactions [76]. Patients with central thalamic pain had slowing of the dominant rhythm and increased theta power, which partially reversed with thalamic surgery [77]. There is also evidence that the dominant posterior rhythm is acetylcholine-mediated as injection of scopolamine induces the slowing of dominant rhythm in healthy controls [78]. DLB is associated with marked degeneration of NBM and pedunculopontine nucleus (PPN) [79], nuclei which provide the majority of cholinergic drive to cortex and thalamus, respectively. Hence, the lack of a normal alpha frequency dominant rhythm in DLB may be a marker of cholinergic dysfunction-associated thalamocortical dysrhythmia [7]. However, thalamocortical dysrhythmia may also be related to noradrenergic and serotoninergic deficiencies as well, being implicated in depression, schizophrenia and obsessive compulsive disorders [76].

Certain symptoms such as cognitive impairment [80], cognitive fluctuations, and visual hallucinations [81,82,83] also correlate with cholinergic deficiency. The slowing of background rhythms, quantified by lower alpha and higher delta appears to correlate with more severe cognitive impairment [51,53,60] and variability of the dominant frequency may be associated with cognitive fluctuations [30,31,34,38]. Whilst dominant frequency variability has been reported to be greater in DLB than AD in some studies [31,34,35,37], the reverse has been reported in others [38,46,50] with, for example, higher theta-alpha variability in AD [38]. In contrast, DLB appears to have greater variability than AD when slower rhythms, particularly delta variability, is examined [30,33]. The degree of cognitive impairment, given its impact on the EEG as noted above, theoretically could be a confounding factor and contributor to inter-group variability; however the majority of studies were well matched in terms of the cognitive scores between DLB and AD patient groups. There may be other drivers for these discrepancies and these may include ascertainment bias (i.e., recruitment from non-homogeneous populations) in the various studies, limited sample sizes and differences in clinical phenotype and concurrent medication use which might influence the EEG (see below).

Overall, limited conclusions can be drawn regarding EEG correlations with impairments in specific cognitive domains, visual hallucinations, neuropsychiatric, motor symptoms and response to cholinesterase inhibitors due to the small number of studies exploring these. One randomised controlled trial reported that donepezil, a cholinesterase inhibitor, increased alpha activity in dementia patients with cognitive fluctuations and reduced dominant frequency variability in those with and without cognitive fluctuations [84]. However, this study was not included in the review as there was no clear distinction if the patients studied had DLB or other dementias. None of the studies have assessed EEG findings in relation to more direct measures of cholinergic function (e.g., cholinergic PET imaging). Evidence that the EEG findings were indicative of cholinergic dysfunction was at best indirect.

Connectivity was abnormal in DLB patients although some reported weaker connectivity than AD while in some the reverse was reported (Table 3). Findings from graph theory analyses consistently reported increased segregation, reduced hubness and a randomised pattern of network in DLB compared to AD. This suggests that DLB is a more severe disconnection syndrome than AD [45,46,47,50,64]. However, due to the heterogeneity of connectivity measures used and the fact that conflicting results had been reported, more studies are needed to explore if the disconnection is, indeed, worse in DLB. Reduced reactivity to eyes opening [63] and intermittent photic stimulation [60] and reduced posterior-to-anterior dPTE suggest the disconnection syndrome affects predominantly posterior regions [48]. Functional connectivity of the basal ganglia, thalamic and cortical networks correlating with longer MS duration in DLB have been implicated as well [55].

EEG characteristics of PDD patients with core DLB symptoms such as visual hallucinations and cognitive fluctuations were similar to those of patients with DLB [34,38,46,51,52]. This supports the hypothesis that PDD and DLB arise from the same spectrum of disease [85]. In many studies, EEG abnormalities in PDD were of intermediate severity, being more marked than in AD but less severely affected than in DLB. A recent systematic review reported similar findings in PD, where EEG slowing, particularly decreased dominant frequency and increased theta power, correlated with cognitive impairment and predicted future cognitive deterioration [23].

Patients with mixed AD/DLB diagnosed clinically or via biomarkers such as CSF tau/AB-42 ratio had intermediate EEG findings as well, being more abnormal than AD but less severe than DLB. Postmortem studies showed that coincident Lewy body pathologies are frequently found in patients clinically diagnosed to have AD [1,2,86]. The presence of hallucinations in patients with AD is strongly predictive of coincident Lewy body pathologies with specificity of 100% in one study [1]. The relatively more severe impairment of executive function compared to memory also predicted coincident Lewy body disease [1].

Although the studies reported moderate to good specificity of EEG features in diagnosing DLB, the interpretation of the findings needs to take into account a number of limitations. Firstly, the EEG definitions and methods between studies are not standardised which makes comparison of results across studies difficult and meta-analysis unfeasible. Furthermore, different EEG acquisition and pre-processing methods and protocols may affect quantitative EEG results even if the same analysis methods were used. Quantitative EEG results may be affected by electrode placement, artefact contamination, band filtering, levels of drowsiness, choice of epochs, age, and medication use [87]. Many studies did not report sufficiently on these factors to reliably account for them. The proportion of participants with DLB taking psychoactive drugs was greater compared to participants with AD in all studies where this information was disclosed. Psychoactive drugs such as benzodiazepines, antipsychotics, antidepressants, anticonvulsants, and dopaminergic agents are commonly prescribed and result in some EEG changes such as enhanced beta activities, increase in theta or delta power, and triphasic waves [88,89]. Secondly, most studies did not comment on whether the EEG assessors were blinded to the clinical diagnosis which constitutes a potential bias although this may be less of a factor in quantitative studies compared to qualitative studies which are dependent on a subjective rating. Apart from one post-mortem study, in most studies the diagnosis was based on clinical criteria which may not always be correct [90]. It is possible that comparator groups of PDD and AD may have mixed disease. Thirdly, none of the studies had prespecified cut-offs for EEG features. Post hoc data-driven analysis may exaggerate test performance, leading to falsely high accuracy [91]. Lastly and crucially, there was lack of external independent validation of the EEG features analyses.

In conclusion, the slowing of the dominant background rhythm detected by visual or quantitative analysis is a sensitive biomarker to differentiate DLB from AD. However, this finding is non-specific and may be accounted for by other diseases, drugs or the level of wakefulness. Quantitative approaches may offer enhanced diagnostic accuracy but there remains a lack of standardisation of EEG acquisition, processing, analysis, and reporting to allow protocols to be replicated and validated externally.

Thus whilst EEG is a highly promising modality, with potentially comparable or indeed better diagnostic characteristics than many other biomarkers and the potential for wider use given its non-invasive nature and low cost (e.g., compared with CSF analysis or neuroimaging), large-scale high quality prospective studies with standardized EEG protocols are required before a more definitive role can be assigned to EEG as a biomarker in DLB.

## Figures and Tables

**Figure 1 diagnostics-10-00616-f001:**
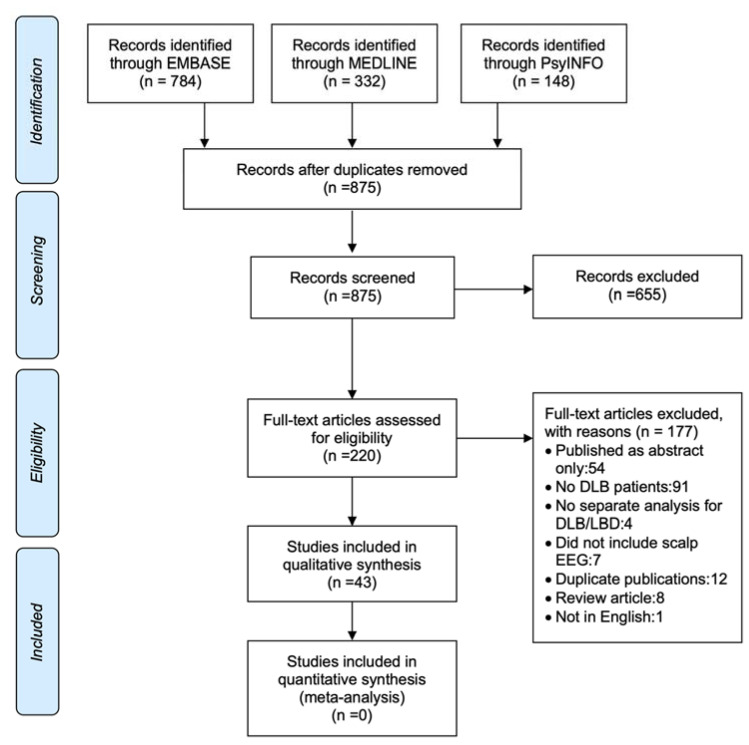
PRISMA flow diagram of study selection.

**Figure 2 diagnostics-10-00616-f002:**
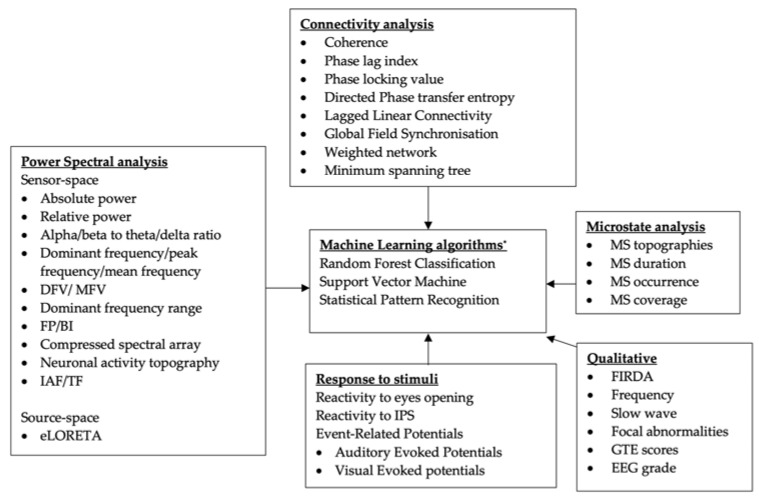
Classification of EEG parameters and analyses in the included studies. * Not a quantitative EEG analysis, may include other biomarkers as features. BI = band inscription; DF = dominant frequency; DFV = dominant frequency variability; eLORETA = exact low resolution brain electromagnetic tomography; FIRDA = frontal intermittent rhythmic delta activities; FP = frequency prevalence; GTE = grand total EEG score; IAF = individual alpha frequency peak; IPS = intermittent photic stimulation; MFV = mean frequency variability; MS, microstate; TF = transition frequency.

**Table 1 diagnostics-10-00616-t001:** Characteristics of included studies.

	Study	No. of Total (DLB) Participants *	Population ^‡^	MMSE ^‡^	Study Design	Single/Multicentre	Diagnosis of DLB	EEG Features	Other Modalities	Overall Quality
1	Briel 1999 [25]	25 (14)	DLB/AD	-	CS	Single	postmortem	Visual EEG assessment	-	moderate
2	Roks 2008 [26]	88 (18)	DLB/AD/SMC	23/22/28	Follow up	Single	DLB 1996	Visual EEG-GTE score	-	high
3	Lee 2015 [27]	83 (29)	DLB/AD	25/24	Follow up	Single	DLB 2005	Visual EEG-GTE score	-	moderate
4	Barber 2000 [28]	38 (18)	DLB/AD	9.4/17.2	Follow up	Single	DLB 1996	Visual EEG Grading	-	moderate
5	Londos 2003 [29]	62 (34)	DLB/AD	-	CS	Multi-	DLB 1996 with postmortem subgroup	Visual EEG Grading	rCBF, neuropathology	low
6	Walker 2000b [30]	40 (15)	DLB/AD/HC	17.3/18.0/29	CC	Single	DLB 1996	Delta variability	-	moderate
7	Walker 2000a [31]	155 (37)	DLB/AD/VaD/HC	17.6/17.8/ 17.8/27.7	CC	Single	DLB 1996 with postmortem subgroup ^§^	Mean frequency variability	-	moderate
8	Kai 2005 [32]	42 (15)	DLB/AD/HC	20/21/-	CC	Single	DLB 1996	Spectral power and coherence	-	moderate
9	Andersson 2008 [33]	138 (20)	DLB/AD/HC	22/23/29	CC	Single	DLB 1996	Spectral power, delta variability, coherence	-	moderate
10	Bonanni 2008 [34]	161 (36)	DLB/PDD/AD/HC	22.8/22.9/ 22.3/28.9	Follow up	Multi-	DLB 1996 (only 2 core feature) with 2-year follow-up confirmation	Spectral power, CSA	SPECT, MRI	high
11	Bonanni 2010 [35]	119 (32)	DLB/AD/HC	22.8/22.1/ 29.0	Follow up	Multi-	DLB 2005	Spectral power, CSA (same patients as Bonanni 2008), P300;	-	moderate
12	Bonanni 2015 [36]	47 (20)	MCI-DLB/MCI-AD/DLB/AD/HC	25.7/25.9/22.7/21.9/28.9	Follow up	Multi-	DLB 2005, MCI with 3-year follow-up	Spectral power, CSA	SPECT, MRI	high
13	Bonanni 2016 [37]	212 (79)	DLB/AD	22.9/22.7	CS	Multi-	DLB 2005	Spectral power, CSA	-	moderate
14	Stylianou 2018 [38]	73 (17)	DLB/PDD/AD/HC	25/23.7/29.2	CC	Single	DLB 2005 and 2017	CSA	-	moderate
15	Snaedal 2012 [39]	654 (52)	LBD ^†^/AD/VaD/sMCI/FTLD/depression	-	CS	Multi-	DLB 2005	Spectral power, coherence SPR	MRI/SPECT/DAT/CSF	low
16	Franciotti 2013 [40]	51 (18)	DLB/AD/HC	20.6/20.4/ 28.9	CC	Multi-	DLB 1996	CSA	MRI/fMRI	moderate
17	Garn 2017 [41]	61 (20)	LBD ^†^/AD/FTLD	21.8/24/23.3	Follow up	Single	DLB 2005	Spectral power, coherence, SVM	-	moderate
18	Engedal 2015 [42]	517 (15)	LBD ^†^/AD/VaD/other Dementia /MCI/SMC/ depression/HC	24.1/23.3/ 22.7/24.3/ 27.1/28.7/ 27.0/28.9	CC	Multi-	DLB 2005	Spectral power, coherence, SPR	-	moderate
19	Ferreira 2016 [43]	411 (15)	LBD ^†^/AD/other Dementia /MCI/SMC/HC	24.1/23.4/ 23.5/27.1/ 28.6/28.9	CC	Multi-	DLB 2005	Spectral power, coherence, SPR	MRI (MTA, PA, GCA-F)/CSF	moderate
20	Colloby 2016 [44]	72 (21)	DLB/AD/HC	22.4/20.8/-	CC	Multi-	DLB 2005	Spectral power, coherence SPR	MRI (MTA)	low
21	van Dellen 2015 [45]	198 (66)	DLB/AD/HC	23/21/28	CC	Multi-	DLB 2005	PLI, MST	MRI (MTA)	moderate
22	Peraza 2018 [46]	89 (25)	DLB/PDD/AD/HC	22.6/22.7/ 20.1/29.1	CC	Single	DLB 2005	Spectral power, PLI, MST	-	moderate
23	Dauwan 2016b [47]	198 (66)	DLB/AD/SMC	23/21/28	CS	Multi-	DLB 2005 and 2017	Visual EEG assessments/grading, Spectral power, MST, PTE, RFC	MRI (MTA, GCA)/CSF	moderate
24	Dauwan 2016a [48]	198 (66)	DLB/AD/HC	23/21/28	CC	Multi-	DLB 2005	dPTE	-	moderate
25	Dauwan 2018 [49]	173 (29)	DLB/AD	23/21	CS	Multi-	DLB 2005	Visual EEG assessment/grading, Spectral power, connectivity, visual, RFC	-	moderate
26	van der Zande 2018 [50]	123 (41)	DLB/AD/mixed DLB-AD	24/22/20	CS	Multi-	DLB 2005	Visual EEG assessment/grading, spectral power, PLI, MST, RFC	DAT/CSF	moderate
27	Babiloni 2017 [51]	158 (34)	DLB/PDD/AD/HC	18.6/18.8/ 18.9/28.7	CC	Multi-	DLB 2005	Spectral power, TF and IAF	-	moderate
28	Babiloni 2018a [52]	158 (34)	DLB/PDD/AD/HC	18.6/18.8/ 18.9/28.7	CC	Multi-	DLB 2005	LLC	-	moderate
29	Babiloni 2018b [53]	83 (23)	MCI-LB/AD-MCI/HC	24.7/25.1/ 28.6	CC	Multi-	DLB 2005/2017	Spectral power, TF and IAF	MRI/FDG-PET/CSF	moderate
30	Babiloni 2019 [54]	83 (23)	MCI-LB/MCI-AD/HC	25.7/25.6/	CC	Multi-	DLB 2005/2017	LLC	MRI/FDG-PET/CSF	moderate
31	Schumacher 2019 [55]	96 (25)	LBD ^†^/AD/HC	23.1/20.7/ 29.2	CC	Single	DLB 2005	Microstates	fMRI	moderate
32	Tanaka 2017 [56]	93 (21)	DLB/AD/HC	-	CC	Multi-	DLB 2005	Spectral power, NAT	-	low
33	Liedorp 2009 [57]	1116 (38)	DLB/AD/VaD/FTLD/MCI/Psych/SMC	-	CS	Multi-	DLB 1996	Visual EEG assessment	-	moderate
34	Kurita 2010 [58]	82 (24)	DLB/PDD/AD/HC	20.1/20.3/ 20.3/28.5	CC	Single	DLB 2005	ERP (visual and auditory)	-	moderate
35	Perriol 2005 [59]	40 (10)	DLB/PDD/AD/HC	-	CC	Single	NA	ERP (auditory)	-	moderate
36	Pugnetti 2010 [60]	42 (10)	DLB/PDD/PD/HC	21.0/18.0/ 25.5/27.7	CC	Single	DLB 1996	GFS	-	moderate
37	Mehraram 2019 [61]	96 (25)	DLB/PDD/AD/HC	22.7/23.4/ 20.2/29.2	CC	Single	DLB 2005 & 2017	Weighted PLI, weighted network measures	-	moderate
38	Aoki 2019 [62]	121 (41)	DLB/HC	21.4/-	CC	Single	DLB 2005	Resting state network	-	moderate
39	Schumacher 2020a [63]	102 (24)	DLB/PDD/AD/HC	23.1/21.6/ 28.8	CC	Single	DLB 2017	Alpha reactivity	MRI	moderate
40	Franciotti 2020 [64]	325 (144)	DLB/AD/HC	22/23/29	CC	Multi	DLB 2005 & 2017	Anterior-posterior dominant frequency	-	moderate
41	Massa 2020 [65]	58 (12)	MCI-LB/PD/MCI-AD/HC	26.9/28.8/ 27.7/29.0	CC	Single	Not mentioned	Alpha/theta ratio	-	moderate
42	van der Zande 2020 [66]	114 (37)	MCI-LB/MCI-AD	27/26	CC	Single	DLB 2017 and MCI-LB criteria	Visual EEG assessment, Spectral power	-	high
43	Schumacher 2020b [67]	106 (39)	MCI-LB/MCI-AD/HC	26.6/26.9/ 28.5	CC	Single	Prodromal DLB 2020	Spectral power		moderate

* Numbers in bracket indicate number of DLB patients ^†^ LBD = Lewy body disease-DLB and PDD were analysed as a combined group. ^‡^ Mean MMSE ordered as per population column. ^§^ Sensitivity of 0.83 and a specificity of 0.91 against neuropathological diagnosis in the first 50 patients. AD = Alzheimer’s disease; CC = case control; CS = case series; CSA = compressed spectral arrays; CSF = cerebrospinal fluid; DAT = dopamine transporter scan; DLB = dementia with Lewy bodies; EEG = electroencephalography; ERP = event-related potentials; FDG-PET = fluorodeoxyglucose-positron emission tomography; fMRI = functional magnetic resonance imaging; FTLD = fronto-temporal lobar degeneration; GCA = global cortical atrophy; GCA-F = global cortical atrophy frontal sub-score; GFS = global field synchronization; GTE = grand total electroencephalography (EEG); HC = healthy controls; IAF = individual alpha frequency peak; LLC = lagged linear connectivity; MCI = mild cognitive impairment; MCI-AD = mild cognitive impairment due to Alzheimer’s disease; MCI-LB = mild cognitive impairment with Lewy bodies; MMSE = Mini-Mental State Examination; MRI = magnetic resonance imaging; MST = minimum spanning tree; MTA = medial temporal atrophy; NA = not available; NAT = neuronal activity tomography; PA = posterior atrophy; PD = Parkinson’s disease; PDD = Parkinson’s disease dementia; PLI = phase lag index; dPTE = directed phase transfer entropy; rCBF = regional cerebral blood flow; RFC = random forest classification; SMC = subjective memory complaints; SPECT = single-photon emission computed tomography; SPR = statistical pattern recognition; SVM = support vector machine; TF = transition frequency; VaD = Vascular Dementia.

**Table 2 diagnostics-10-00616-t002:** Comparison of relative band power between dementia with Lewy bodies and Alzheimer’s disease.

Relative Power	DLB	AD
Beta	6–19%	15–25%
Alpha	11–29%	24–35%
Theta	28–40%	11–23%
Delta	34–44%	28–32%
Theta/alpha ratio	0.51–1.09	0.40–0.48

**Table 3 diagnostics-10-00616-t003:** Summary of studies on EEG connectivity measures in DLB.

Studies	Band	Measure	Result
Andersson 2008 [33]	Beta	Coherence	AD > DLB
Dauwan 2016a [48]	Beta	PTE	AD > DLB
Mehraram 2019 [61]	Beta	WPLI	AD > DLB
Kai 2005 [32]	Beta	Coherence	AD < DLB
Andersson 2008 [33]	Alpha	Coherence	AD > DLB
van Dellen 2015 [45]	Alpha	PLI	AD > DLB
Dauwan 2018 [49]	Alpha	PLI	AD > DLB
van der Zande 2018 [50]	Alpha	PLI	AD > DLB
Kai 2005 [32]	Alpha	Coherence	AD < DLB
Dauwan 2016a [48]	Alpha	PTE	AD < DLB
Babiloni 2018a [52]	Alpha	LLC	AD < DLB/PDD
Andersson 2008 [33]; Kai 2005 [32]	Theta	Coherence	AD < DLB
Babiloni 2018a [52]	Delta	LLC	AD < DLB/PDD

AD = Alzheimer’s disease; DLB = dementia with Lewy bodies; LLC = lagged linear connectivity; PDD = Parkinson’s disease dementia; PLI = phase lag index, PTE = phase transfer entropy; WPLI = weighted phase lag index. AD > DLB indicates weaker connectivity in DLB compared to AD; AD < DLB indicates weaker connectivity in AD compared to DLB.

**Table 4 diagnostics-10-00616-t004:** Diagnostic accuracy of specific EEG features.

EEG Features	Studies	Sensitivity	Specificity	Accuracy	AUROC
DLB vs. AD					
EEG severity grade	Barber 2000 [28]; Londos 2003 [29]; van der Zande 2018 [50]	97%	100%	99%	-
GTE	Roks 2008 [26]; Lee 2015 [27]	72–79%	76–85%	-	0.78–0.90
Occipital alpha power	Babiloni 2017 [51]; Babiloni 2018b [53]	65–78%	67–74%	70–73%	0.72–0.75
Parietal delta power	Babiloni 2018b [53]	78%	67%	73%	0.72
Delta SD	Andersson 2008 [33]	75%	80%	-	-
Theta FP + theta power + theta-alpha DFV	Stylianou 2018 [38]	92%	83%	-	0.94
CSA pattern	Bonanni 2015 [36]	~100%	~100%	~100%	-
PLI beta band	Dauwan 2018 [49]	93%	97%	95%	-
MST-PLI	Peraza 2018 [46]	80%	85%		0.86
Weighted PLI network	Mehraram 2019 [61]	47%	100%	66%	0.78
P300- reversed amplitude distribution gradients	Bonanni 2010 [35]	70%	97%	-	-
Machine learning algorithms	Garn 2017 [41]; Mehraram 2019 [61]; Snaedal 2012 [39]; Engedal 2015 [42]; Ferreira 2016 [43]; Colloby 2016 [44]	76–100%	77–100%	66–100%	0.78–0.93
MCI-LB vs. MCI-AD *					
EEG severity grade > 2	van der Zande 2020 [66]	-	-	-	0.76
Diffuse abnormalities	van der Zande 2020 [66]	-	-	-	0.84
Peak/dominant frequency	van der Zande 2020 [66]; Schumacher 2020b [67]	51%	86%	-	0.70–0.89
Beta power	van der Zande 2020 [66]; Schumacher 2020b [67]	61%	81%	-	0.71–0.91
Alpha power	van der Zande 2020 [66]; Schumacher 2020b [67]	41%	97%	-	0.66–0.85
Pre-alpha power	Schumacher 2020b [67]	56%	83%	-	0.68
Theta power	van der Zande 2020 [66]; Schumacher 2020b [67]	33%	89%	-	0.60–0.94
Delta power	van der Zande 2020 [66]; Schumacher 2020b [67]	23%	89%	-	0.54–0.55
Theta/alpha ratio	van der Zande 2020 [66]; Schumacher 2020b [67]	49%	83%	-	0.64–0.92
Theta/alpha ratio + alpha1power + FIRDA	van der Zande 2020 [66]	-	-	-	0.97

AUROC = area under receiver operating characteristics curve; CSA = combined spectral arrays; DFV = dominant frequency variability; FIRDA = frontal intermittent rhythmic delta activities; FP = frequency prevalence’ GTE = grand total EEG; MCI-AD = mild cognitive impairment due to Alzheimer’s disease; MCI-LB = mild cognitive impairment with Lewy bodies; MST = minimum spanning tree; PLI = phase lag index; SD = standard deviation. * Sensitivities and specificities for MCI-LB vs. MCI-AD were all reported by Schumacher 2020b; AUROC reported by both Schumacher 2020b and van der Zande 2020.

**Table 5 diagnostics-10-00616-t005:** Correlation of EEG features with domains of cognitive function.

Study	Test	EEG Features	Correlation Coefficient (rs)
	Frontal lobe		
Bonanni 2010 [35]	FAB	P300 latency on auditory ERP	−0.6
Bonanni 2010 [35]	FAB	P300 amplitude on auditory ERP	−0.5
	Attention/concentration		
Aoki 2019 [62]	revised Wechsler Memory Scale	Occipital alpha activity	0.45
	Executive function		
van Dellen 2015 [45]	TMT-B	connectivity strength on MST	0.31
Dauwan 2016a [48]	TMT-B	mean PTE gradient in posterior brain regions in the beta band	−0.37
Mehraram 2019 [61]	FAS verbal fluency	node degree in alpha range on weighted PLI	−0.46
Mehraram 2019 [61]	FAS verbal fluency	average path length in alpha range on weighted PLI	0.44
Aoki 2019 [62]	Verbal fluency	Occipital alpha activity	0.34
Aoki 2019 [62]	Verbal fluency	memory perception network	0.36
Aoki 2019 [62]	Verbal fluency	sensorimotor network	−0.44
	Language		
Mehraram 2019 [61]	Animal naming	clustering coefficient of weighted PLI network	−0.44
	Visual perception		
van Dellen 2015 [45]	visual association test	Leaf fraction-MST	0.29
van Dellen 2015 [45]	visual association test	tree hierarchy-MST	0.29
van Dellen 2015 [45]	visual association test	connectivity strength-MST	0.33
Aoki 2019 [62]	shape discrimination	Occipital alpha activity	−0.46
Aoki 2019 [62]	shape discrimination	memory perception network	−0.41

FAB = frontal assessment battery; MST = minimum spanning tree; TMT-B = trails making test-B.

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
