# Peer review of "The Role of EEG in the Diagnosis, Prognosis and Clinical Correlations of Dementia with Lewy Bodies—A Systematic Review"

_diagnostics, 2020, doi:10.3390/diagnostics10090616_

Round 1
Reviewer 1 Report
This systematic review of all EEG studies in DLB compared to AD and other dementias is a very important task.
Specifically:1. It has been very carefully designed to serve the primary purpose to clarify the diagnostic role of EEG in DLB compared to AD and other dementias.2. All the data of the qualitative and quantitative assessment of the EEG have been correctly evaluated and at the same time explaining the parameters of the quantitative analysis.3. With regard to the explanation of the conclusions, I would like to point out the following:
At the point of discussion, where it refers to the variability of EEG findings in both AD and DLB, it would be appropriate to mention, among other things, the grade of severity of AD as an additional factor that may contribute to this variability.
It is known that the mild diffuse slowing observed in the early stages of Alzheimer's disease may worsen in the later stages of the disease:
It would be useful, if referring to the evaluated studies, to highlight the EEG findings in moderate Alzheimer's stages relative to corresponding DLB stages.
That is, in addition to the general report that EEG findings are deteriorating in relation to the severity of dementia, it would be useful to clarify whether the progression of the severity of dementias is related to quality characteristics.
Also in the discussion and specifically in the assessment of conclusions regarding connectivity:In my opinion, the pathological EEG findings of DLB are overemphasized compared to Alzheimer's disease, while at the same time the limitations of the study are analyzed very correctly.
In conclusion, I think it would be useful in the discussion to refer to the relationship between the progression of dementias and the maintenance of quality EEG characteristics.
Also, the assessment of EEG findings in relation to connectivity should be made with caution due to the many limitations of the study.
Reviewer 2 Report
This study reviewed the role of electroencephalography (EEG) in the diagnosis of dementia with Lewy bodies (DLB). This review is very important because the different diagnosis between DLB and other diseases is often difficult. I have some comments, which are as follows:
- In the 2017 DLB consortium diagnostic criteria, the posterior slow-wave activity on resting EEG is a supportive biomarker. However, this review showed that DLB patients had EEG abnormalities not only in the posterior region, but also in the anterior and temporal regions. Could the EEG abnormalities in the anterior and temporal regions be supportive biomarkers of DLB?
- Authors mentioned in the introduction that EEG abnormalities are often non-specific to DLB. In clinical setting, it is often difficult to distinguish between DLB and psychiatric diseases including depression. Are there any EEG studies reported the different diagnosis between DLB and psychiatric diseases?
Round 2
Reviewer 2 Report
I have reviewed the revised manuscript which incorporates my recommendations. I think the manuscript is excellent and recommend it highly for publication.